# Integration of Nanometer-Range Label-to-Label Distances and Their Distributions into Modelling Approaches

**DOI:** 10.3390/biom12101369

**Published:** 2022-09-25

**Authors:** Gunnar Jeschke

**Affiliations:** Department of Chemistry and Applied Biosciences, ETH Zürich, Vladimir-Prelog-Weg 2, CH-8093 Zürich, Switzerland; gjeschke@ethz.ch

**Keywords:** EPR spectroscopy, double electron electron resonance, FRET, ensemble model, intrinsic disorder, structural biology, site-directed spin labelling, molecular force fields

## Abstract

Labelling techniques such as electron paramagnetic resonance spectroscopy and single-molecule fluorescence resonance energy transfer, allow access to distances in the range of tens of angstroms, corresponding to the size of proteins and small to medium-sized protein complexes. Such measurements do not require long-range ordering and are therefore applicable to systems with partial disorder. Data from spin-label-based measurements can be processed into distance distributions that provide information about the extent of such disorder. Using such information in modelling presents several challenges, including a small number of restraints, the influence of the label itself on the measured distance and distribution width, and balancing the fitting quality of the long-range restraints with the fitting quality of other restraint subsets. Starting with general considerations about integrative and hybrid structural modelling, this review provides an overview of recent approaches to these problems and identifies where further progress is needed.

## 1. Introduction

In the early decades, structural biology followed Anfinsen’s thermodynamic hypothesis [1] that, under normal physiological conditions, the peptide sequence encodes a single deep minimum in the potential energy surface. This minimum corresponds to a single protein conformation. So the goal was to determine a unique three-dimensional structure with the best possible resolution. X-ray diffraction of protein crystals and later NMR spectroscopy of proteins in solution became the main approaches of structural biology. After a “resolution revolution”, cryo-electron microscopy could also provide structures at atomic resolution. Recently, solid-state NMR has been developed into another approach to this problem.

At the turn of the millennium, it became clear that a substantial fraction of proteins and protein domains are intrinsically disordered in functional states [2,3,4,5]. In 2015, Peng et al. quipped that these exceptions are exceptionally abundant in all domains of life, with eukaryotic proteomes being about 20% disordered [6]. Their analysis found that disordered regions exhibit some sequence conservation, but less conservation than structured regions and that disorder is enriched, among else, in transcription, translation, and RNA splicing. Disorder is involved in the interaction of viruses with other organisms and especially in their RNA binding [6]. In many cases, the disorder of proteins or their domains is not complete in the sense that the macromolecules could be described by a random coil. Instead, proteins populate a continuum of order and disorder, and the extent of disorder can be heterogeneous even within the same domain [7]. Proteins featuring disorder are underrepresented in structural databases [6]. They may be overrepresented among systems studied using site-directed spin labelling (SDSL) and electron paramagnetic resonance (EPR) spectroscopy, as this approach is often considered a last resort for systems that defy full characterisation by better established approaches. By measuring distance distributions through the combination of SDSL with double electron resonance (DEER), many EPR studies have thus found that a description in terms of a single conformer is inadequate [8]. This raises the question of how to build reliable ensemble models of proteins and protein complexes.

About two decades of research have shown that, in general, no single experimental approach is able to provide a sufficient set of restraints for building an ensemble model that fully accounts for the residual order of a partially disordered system. This, along with the same realisation in structure determination of large molecular machines [9,10], has led to approaches that integrate experimental restraints from different techniques. Such approaches are referred to as integrative in this review. In a few cases, even the data collected by several experimental techniques may not suffice for generating a sufficiently detailed model. One can then resort to including information from knowledge bases, such as molecular force fields [11,12], fragment libraries [13,14], or structure predictions based on homology and coevolution [15]. Such approaches are referred to as hybrid in this review.

An overview of all relevant developments in ensemble modelling with integrative and hybrid approaches would probably require an entire book. Instead, we will here approach the topic from the perspective of nanometer-scale distance measurements, such as those that can be made between labels on a protein or nucleic acid using DEER EPR [16] or single-molecule fluorescence resonance energy transfer (smFRET) [17]. In particular, we will focus on the distance *distribution* information obtained with DEER EPR. This information is unique in that it provides information about the width of the ensemble and thus the extent of disorder. Our focus means that we cannot do full justice to the parallel developments in smFRET. Nevertheless, we discuss the complementarity between smFRET and DEER EPR as well as some methodological analogies and differences.

This review is organised as follows. First, we explain the goal of ensemble modelling and discuss on a few examples how such models can help in understanding protein function. Second, we consider the general flow of information in ensemble modelling through integrative or hybrid approaches. Then, we approach the problem from the perspective of statistical thermodynamics to provide a solid foundation for the discussion of the continuum of order and disorder. In doing so, we point out that the energy hypersurfaces underlying protein function, experimental measurements, and molecular force fields are subtly different. We critically discuss different approaches to generating a raw ensemble and reweighting the ensemble. Then we consider the specifics of label-based restraints. In particular, we address the need to represent labels in the model and discuss the available approaches for that. We then turn to distribution restraints and discuss the best ways to implement them in raw ensemble generation and ensemble reweighting. We conclude with thoughts on necessary future developments, in particular the separation of ensemble width and ensemble uncertainty.

## 2. Ensemble Models and Biological Function

Ensemble modelling aims at a three-dimensional representation of the structure of partially disordered systems. Individual conformers can be represented at atomic resolution or by coarse-grained models. In the following, atomic resolution is implicitly assumed, but coarse-grained modelling is done by close analogy. While such ensemble models can be used to study the polymer physics aspects of proteins and protein complexes [18], their greatest utility is in visually and intuitively understanding the functional aspects of the systems. In this context, distance distribution information can reveal how distribution of structure is related to function. In an early example, bimodal distance distributions between spin-labelled sites of the non-coding RNA RsmZ upon binding to the protein RsmE showed that RsmZ can form two major conformers of a complex with three copies of RsmE [19]. RsmE in turn represses translation initiation. The distributed three-dimensional structure explains how the small RNA RsmZ sequesters RsmE and thus de-represses translation initiation. In this case, distance distributions between spin-labelled sites were essential to obtain a structural model at all. However, the approach used in this study modelled narrowly distributed ensembles for the two major conformers separately and was thus not easily extensible to continuous distribution of structure.

In another early example, a small ensemble of six major conformations of the endosomal complex required for sorting and transport ESCRT-I was obtained by integrating FRET, DEER, and SAXS data [20]. The protein is involved in the formation of membrane buds on several processes in yeast cells. In this study, distance distributions provided direct evidence that the N-terminal predicted helix, the ubiquitin E2 variant domain, and the C-terminal domain were all conformationally heterogeneous relative to the core. Unexpectedly, about 50% of the protein was found in a closed form. Clearly, the presence of, both, closed and open conformations of ESCRT-I in solution must be considered when elucidating how ESCRT-I induces membrane buds. The authors pointed out that structure of ESCRT-I should not be understood in terms of only six discrete conformers, but rather as a continuum of conformers. This in turn indicated a need for methodological advances in ensemble modelling based on labelling techniques.

In a recent example, ensemble modelling based on distance distribution restraints and a small-angle x-ray scattering (SAXS) curve revealed the location of the glycine-rich C-terminal intrinsically disordered domain of heterogeneous nuclear ribonucleoprotein A1 (hnRNP A1) that is involved in alternative splicing regulation, mRNA export from the nucleus to the cytoplasm, and segregation of RNA into stress granules [21]. The ensemble model of the free, dispersed form of hnRNP A1 revealed that the intrinsically disordered domain is rather compact and contacts and shields the RNA-binding interface of the structured domains. This offers an explanation for how RNA binding induces liquid–liquid phase separation of the full-length protein. When RNA binds, the intrinsically disordered domain must be replaced, so that sections are exposed to an aqueous environment that have a preference for intra-protein interaction.

## 3. Information Flow in Ensemble Modelling

Integrative modelling of biomolecular structures is often performed as ensemble modelling, where the width of the ensemble is related to both the uncertainty of the model and the disorder of the modelled system. Information flow and key modules (Figure 1) are common to different approaches to the problem. The experimental phase of the project provides primary data, the interpretation of which depends on further information about the sample and the modalities of the measurements. The first stage of the modelling pipeline analyses this data and converts it into structural restraints. In many cases, this includes conversion to spatial restraints, such as distances, angles, and torsion angles. In some cases, however, it is preferable to restrain the model with standardised, experiment-specific data, such as phase- and zero-time-corrected primary DEER data (see Section 7.1).

The second stage of the pipeline generates a raw ensemble of conformers of the system, where we use the term conformer for both the conformation of individual macromolecules and their mutual arrangement (docking). The critical aspect of this stage is sufficient sampling of the conformer space to avoid missing valid solutions. Such sampling is based on statistics about possible conformations and is more efficient the better these statistics are known. Sampling cannot be improved by restraints that can only be simulated with a complete ensemble. However, it can be improved by distribution restraints that provide information about the probability of occurrence of individual conformers. In hybrid modelling, sampling is improved by relying on a molecular force field, often by performing a molecular dynamics (MD) simulation.

The raw ensemble of conformers output from the second stage of the modelling pipeline is not expected to provide the best fit of all restraints. Therefore, the third stage of the pipeline fits the ensemble to the restraints by selecting a subset of conformers from the raw ensemble or by assigning weights to the conformers. These two approaches can be combined by interpreting the weights as populations of representative conformers and discarding conformers whose populations are below a threshold. This stage yields a representative ensemble whose optimal size is discussed in Section 4.1 based on thermodynamics and resolution arguments.

It is important to note that the third stage always yields an ensemble model, even if the experimental data are insufficient or inconsistent. The model must therefore be validated in the fourth stage of the pipeline, for example by systematically omitting restraints in bootstrapping or jacknife resampling [18] approaches. Alternatively, the model can be used to predict previously unused restraints or the outcome of new biophysical or biochemical experiments. Validation may also include visual inspection of the conformers and their interpretation in terms of the biological function of the system.

Finally, the validated representative ensemble model is deposited together with metadata in a repository from which it can be freely downloaded. The most popular repository for biomolecular structures, the Protein Data Bank (PDB), does not yet accept models obtained by integrative or hybrid modeling. However, there is a development version PDB-Dev that will be integrated into the PDB once the modalities for structure submission and curation are established [22,23]. Ensembles of intrinsically disordered proteins (IDPs) that do not meet PDB-Dev requirements can be deposited in the Protein Ensemble Database [24].

## 4. Thermodynamic Description of Biomolecular Structure

All approaches to biomolecular ensemble modeling are based on concepts of statistical thermodynamics. Although these concepts provide a convenient common framework for discussing the various approaches, they are rarely explained explicitly. Here we provide such a discussion because the special role of long-range and distribution restraints follows from it.

### 4.1. Relation between Energy Hypersurface and Ensemble Representation

We make the usual assumption that the structure of a biomolecular system can be described within the framework of equilibrium thermodynamics. For simplicity, we discuss concepts for a system consisting of a single peptide chain whose bond lengths and angles are fixed within the uncertainty of the structure. The structure of a single chain is thus completely characterized by a vector $\vec{\chi }$ of backbone and side-chain torsion angles. The concepts can be extended by analogy to nucleic acids and complexes of biomolecules. At any given time, the microstate of a large number of copies of this system is described by the set of vectors ${\vec{\chi }}_{m}$ of all *M* molecules with indices *m*. This microstate is dynamic on a sub-nanosecond time scale. The full information on the microstate is not experimentally accessible. Instead, we must settle for a description in terms of a macrostate that is consistent with all experimental observations and allows prediction of all observables. The macrostate corresponds to an ensemble average over all molecules observed in an experiment and an average over the observation time. We assume the equivalence of ensemble and time average, i.e., an ergodic system.

Under these assumptions, the distribution of $\vec{\chi }$ is a Boltzmann distribution that can be calculated from the energy hypersurface $E(\vec{\chi })$ and the temperature *T*. Even with a discretisation of the torsion angles, the number of potential conformers is far too large to represent the structure by the totality of conformers and associated populations. Instead, we opt for a set of representative conformers and their populations. Conceptually, this description corresponds to a clustering of the ensemble of all conformers that appear in the sample and during the observation period. Each cluster is represented by its central conformer, which is structurally the least different from all other conformers in the cluster. In such clustering we do not want to lose resolution. The smallest number of clusters that satisfies this condition is related to the experimental resolution. We require that the structure of conformers in each cluster does not differ by more than the spatial resolution of our experimental observations. For a given spatial resolution, the smallest number of representative conformers (clusters) is determined by the shape of the energy hypersurface. For simplicity, we discuss this relation for a projection of the hypersurface onto a dimension with coordinate ξ (Figure 2).

The simplest case (Figure 2a) corresponds to the thermodynamic hypothesis of Anfinsen [1], which assumes a single minimum of the hypersurface lower than the rest of the hypersurface by a large multiple of the thermal energy $k_{B}T$. In this limit, only the single conformer corresponding to this minimum is substantially populated. Such a complete enthalpic control of the structure is always only an approximation to the native state. For example, surface-exposed side chains typically populate multiple rotameric states, and in almost all proteins, short terminal segments exhibit some disorder. However, for small globular proteins or globular domains of proteins, this approximation is often sufficient to understand most of the structure-function relationship. In this limiting case, the system can be represented and interpreted in atomistic detail by a single conformer.

In general, the function of biomolecules depends on binding events or changes of state that are accompanied by structural changes. Insofar as the energy hypersurface depends on the concentrations of all components of a cell or organelle, such restructuring can be explained by a change in the energy hypersurface. Often, however, the minima corresponding to all functional structures of a system exist on all relevant hypersurfaces, only their relative energies change when a state changes. When the energy difference between such minima is only a small multiple of $k_{B}T$, more than one state is substantially populated, as shown in Figure 2b for a system with two states. Representing the macrostate of such a system only by the lowest energy conformer leads to erroneous predictions of observables and loss of information about the structure-function relationship.

The situation is most complicated when the energy landscape is rugged on the order of $k_{B}T$ (Figure 2c). Such systems combine structural preferences with high flexibility. The ensemble of conformers is broad, and preferences for subsets of conformers shift with small changes in external conditions. Presumably, this class of systems is well suited for regulation and promiscuous binding. For a correct description of the macrostate, an extensive ensemble of representative conformers is required.

The situation simplifies again when flexibility, and hence disorder, increases further. The ensemble encoded by an energy hypersurface that is virtually flat on the order of $k_{B}T$ approaches a random polymer coil. This limit of complete entropic control is found in denatured proteins [25] and in some intrinsically disordered proteins and protein domains [26]. When the state of a system is experimentally indistinguishable from such complete disorder, two parameters of a self-avoiding random walk chain model are sufficient to describe the macrostate.

### 4.2. Integrative and Hybrid Structural Biology

How can we derive the best description of the macrostate from experimental data? In the Anfinsen limit (Figure 2a), it is often sufficient to collect data using a single experimental technique such as X-ray diffraction, cryo-electron microscopy (cryo-EM) or NMR spectroscopy and interpret them in terms of atomic coordinates for a single conformer. In favourable cases, NMR [27,28] or cryo-EM data [29] also suffice for inferring the macrostate of systems that populate a small number of states (Figure 2b). However, many systems of interest either cannot be captured by one of these techniques or require an ensemble description. In this situation, the set of experimental structural restraints that can be obtained with a single technique is often too sparse. Sparsity means that the number of restraints is not sufficient to specify the macrostate at a resolution close to the experimental resolution of the technique. In such a situation, it is necessary to integrate restraints from different experimental techniques. In this sense, structure determination by NMR can be considered as an integrative approach [30], because it usually combines diverse information from multiple experiments.

In the thermodynamic framework, the experimental restraints are implicitly used to reconstruct the energy hypersurface that determines the macrostate. In general, the experimental data specify spatial restraints, each related to the distribution P(ξ) of a particular spatial variable ξ. The distribution P(ξ) can be calculated from the energy hypersurface via the Boltzmann distribution and corresponds to a projection onto the ξ axis. Vice versa, the distribution P(ξ) defines a potential of mean force along ξ (Figure 3). If the number of available projections is sufficiently large, the implicit reconstruction of the energy hypersurface can succeed with sufficient detail. If the number of projections is too small, the macrostate can still be characterized in sufficient detail by augmenting the experimental information with a molecular force field or other information derived from a knowledge base. A force field defines its own complete energy hypersurface and thus its own macrostate. Experimental restraints are used to correct for inadequacies of the force field (see below).

The concept of implicit reconstruction of the energy hypersurface leads to a classification of experimental restraints. Most methods provide a list of single-valued restraints, where each restraint corresponds to a mean value of $P(\xi_{i})$ for a given spatial parameter $\xi_{i}$. Such mean-value restraints do not contain direct information about the width of the conformer ensemble. Scattering curves from small-angle X-ray or neutron scattering experiments (SAXS and SANS) provide global distribution restraints, where ξ corresponds to molecular shape rather than a geometric parameter defined between sites in the molecule. SDSL combined with DEER EPR measurements provides distribution restraints $P(r_{ij})$ for pairs of sites (i,j) corresponding to a potential of mean force between these sites. The agreement of mean-value restraints and global distributional restraints with experiments can only be evaluated at the ensemble level. In contrast, individual conformers can be tested for their consistency with site-to-site distribution restraints.

Characterisation of the distribution of conformers by label-based methods combined with distance distribution measurements is illustrated in Figure 4 for the RNA-binding protein hnRNP A1 that contains both an ordered and a largely disordered domain. When a label is attached to a site in the ordered domain, its spatial position (green) is still somewhat distributed due to conformational ambiguity of the linker by which it is attached. However, this distribution is much narrower than the spatial distribution that results for attachment at a site in the disordered domain (orangered). The distance distribution for such a label pair contains information on the distribution of conformers that would be largely lost when considering only the distance between the mean label positions.

### 4.3. Uncertainty and Bias of Energy Hypersurfaces

The aim is to obtain a macrostate description of the native state of the biomolecular system in a particular cell state. This cell state is characterised by the expression level of the protein under consideration and its potential binding partners or a certain concentration range of low molecular weight ligands, as well as by a physiological temperature range. It corresponds to the native energy hypersurface indicated by the black line in Figure 2a. The energy hypersurface is subject to some uncertainty, indicated by the semitransparent band. The uncertainty arises, for example, from the concentration ranges of other constituents in the same cell state. For consistency, we must also specify a reference temperature. The temperature variation then corresponds to a variation of the energy hypersurface with respect to a reference hypersurface. Conceptually, this variation results from the calculation of the Boltzmann distribution of conformers at the current temperature and the subsequent Boltzmann inversion at the reference temperature. The uncertainty of the native energy hypersurface leads to some uncertainty of the macrostate.

Note that the energy hypersurface underlying an experimental measurement generally differs from the native energy hypersurface (red and orange-red lines in Figure 2a). This difference or bias results from the different composition of the sample compared to the cellular environment of the system and, in many cases, from the difference between the physiological temperature and the temperature of the measurement. The hypersurface underlying an experiment is subject to some uncertainty arising from the uncertainty in the compositional parameters and temperature. In labelling techniques, additional uncertainty is introduced by the label, the presence of which can affect the conformational preference of the macromolecular backbone. Although it appears that this introduces bias rather than uncertainty, it is better to think of it as the latter, since different restraints in the same set are obtained with labels at different locations. Biases and uncertainties in the energy hypersurfaces are different for different techniques because each technique has different sample and temperature requirements. An integrative approach determines a macrostate corresponding to an average of the hypersurfaces underlying each experiment.

A molecular force field determines an energy hypersurface without any uncertainty (blue line in Figure 2a) and with three sources of bias. Description bias results from simplifying the molecular physics in the form of the force field, e.g., by using harmonic potentials or neglecting polarisability. Parameterisation bias results from imperfect determination of force field parameters. Compositional bias results from differences between the simulated and native systems, e.g., a much higher effective protein concentration in the simulations. The bias of the molecular force field is often tolerable in the Anfinsen limit, where it causes a distortion of the macrostate smaller than or comparable to the experimental spatial resolution. The bias has particularly severe consequences for energy hypersurfaces that are rugged on the scale of thermal energy $k_{B}T$, since in this region even a small bias leads to large shifts in the conformer distribution. Molecular force fields may also fail in predicting the radius of gyration in the limit of complete disorder [32,33]. The latter problem can be addressed by reparameterization (see [33,34]). However, the effectiveness of such reparameterization in reducing biases in other regimes is unknown. Improving force fields for intrinsically disordered proteins is an active area of research where it has been observed that such improvement can degrade accuracy in describing ordered domains [35]. Although a force field adequate for both folded and disordered protein states has been developed [36], description of systems corresponding to rugged energy hypersurfaces remains a challenge [37]. In general, molecular force fields are expected to provide an accurate description of local structure, especially in ordered regions. Some of the associated approximations reduce their predictive power for long-range interactions. Distance restraints in the range of 20 Å upwards are expected to compensate for this limitation and could therefore be a particularly suitable complement to force fields.

## 5. Integrative and Hybrid Ensemble Modelling

The construction of a representative ensemble from experimental restraints, possibly augmented by a molecular force field, can be done in different ways implemented fully or in part in a number of software packages (Table 1). In all approaches, the conformational space is sampled and a large ensemble of conformational candidates is reduced to a smaller representative ensemble. The information to be integrated can be very diverse, as is illustrated in Figure 5. Because of bias of the individual pieces of information, the total body of information in general is not fully consistent. Deviations between the consensus integrative ensemble model and individual pieces of information must be balanced, which is complicated by the fact that uncertainty of the information is often not fully known. Further difficulties arise in estimating uncertainty of the representative ensemble from uncertainty of the input information and in separating natural distribution of 3D structure from uncertainty of the model.

### 5.1. Monte Carlo Sampling and Scoring

In principle, we would prefer to sample the conformational space exhaustively. A simple consideration shows that this is not feasible. Already without considering side-chain rotamers, the number of conformers for a peptide chain with *N* residues is $\prod_{n=1}^{N}b_{n}$, where the $b_{n}$ is the number of backbone rotamers per residue. Even if $b_{n}$ is reduced to the number of local minima in residue-specific Ramachandran diagrams [38] ($b_{n}=2\ldots 6)$, the total number of conformers far exceeds the capabilities of computers already at a chain length of N=30. The problem is solved by Monte Carlo generation of a large ensemble of conformers, assuming that this ensemble covers the conformational space sufficiently well to derive a valid representation of the macrostate. Site-to-site distance distribution restraints can be evaluated at the conformer level and can thus improve sampling efficiency [39]. In any case, sampling efficiency is improved by using residue-specific Ramachandran angle statistics, such as those implemented in Flexible Meccano [40] or MMMx [41].

The representative ensemble is then generated by selecting conformers from the raw ensemble and assigning populations. It is possible to assign uniform populations. However, non-uniform populations provide a macrostate description with the same predictive quality at a smaller ensemble size. The selection of representative ensembles can be based on a genetic algorithm, as implemented in ASTEROIDS [42], or on fitting the population vector to experimental restraints, as implemented in MMMx [41]. Provided that the sampling of the conformational space is sufficient, equivalent ensemble descriptions are expected regardless of the approach used for generating the raw ensemble. The Integrative Modeling Platform (IMP) provides a number of different optimizers for selecting a representative ensemble [43].

### 5.2. Molecular Dynamics Approaches

Hybrid modeling using a molecular force field is usually implemented by molecular dynamics (MD) simulations. Such approaches aim to compensate for bias of the force field by adding a counter-bias derived from experimental restraints [44,45]. Small or medium-sized ensembles can be obtained in this way by replica MD simulations [46,47,48]. Alternatively, larger raw ensembles obtained from “unbiased” MD simulations can be refined, which is often referred to as ensemble reweighting. Both types of approaches can be based on maximum entropy considerations [44] or on Bayes’ theorem [49]. Bayes’ theorem in this context states that the conditional probability P(C|D) of a conformer *C* given experimental data *D* is equal to the product of the conditional probability P(D|C) of occurrence of these data for the conformer and the prior probability $P_{0}\left(C\right)$ of occurrence of the conformer in the absence of experimental information. The calculation of P(D|C) is usually straightforward and allows for the consideration of experimental uncertainties. Difficulties can arise in the presence of unknown contributions to experimental uncertainty, such as systematic errors, and unknown uncertainty in the physical model used in predicting experimental data from the structure. The prior is usually chosen as the result of an unrestrained MD simulation with a molecular force field considered appropriate for the system under study [50]. Insofar as different force fields differ in their energy hypersurface for disordered or partially ordered proteins, the outcome of Bayesian approaches will depend on the chosen force field even in the limit of complete sampling of the conformational space. To the best of our knowledge, this problem has not yet been investigated systematically.

PLUMED-ISDB is a toolbox that provides a computational framework for Bayesian metainterference applied to integrative structural biology [51]. Bayesian refinement of ensembles (BioEn) [46] is applicable to raw ensembles regardless of their origin, but is most easily connected to the thermodynamic framework for MD simulations. The prior is then defined by the energy hypersurface of the molecular force field, which is known to be biased with respect to the native energy hypersurface. Directly using probabilities from an “unbiased” MD simulation (i.e., a simulation that includes the bias of the force field with respect to the native energy hypersurface) as the prior would leave some of the bias of the force field uncorrected. To mitigate this problem, the BioEn approach corrects the prior via a relative entropy formalism by assigning a confidence level θ>0 to the force field. Although θ also cannot be derived from first principles unless the magnitude of the force field distortion is known, refinement can be performed for multiple values of θ and an optimal value selected by an L-curve criterion (for details, see [46]).

Alternatively, experimental restraints can be introduced into MD simulations as ensemble-averaged harmonic restraints [52,53]. It has been argued that in this way a single ensemble-averaged restraint affects not only the mean conformation but also inappropriately the variance of the conformation, i.e., the ensemble width, whereas this problem can be avoided in maximum entropy approaches [44]. Therefore, maximum entropy or Bayesian approaches provide a more realistic estimate of ensemble width than mean-value harmonic restraints, assuming that the underlying force field provides a realistic estimate of that width. If distributional restraints are available, the quality of the ensemble width estimate can be tested. The state of development of force fields for difficult systems, approaches to reweighting, and experimental bias in MD simulations have recently been discussed [54].

### 5.3. Separation of Ensemble Width from Uncertainty

Uncertainties in experimental data, in their prediction from structure, and in the energy hypersurface lead to uncertainties in the conformation, even in the Anfinsen limit. This effect is different from the true distribution of the conformation, which results from the finite width of the global minimum of the energy hypersurface and from the presence of local minima within a few $k_{B}T$. The predictive power of a structural model is improved if the width of the ensemble due to the conformational distribution can be separated from the conformational uncertainty. Conceptually, this requires a representation in terms of a super-ensemble whose members are ensembles that are all consistent with experimental data. The variability of the conformation within each ensemble is a feature of the biomolecular system that may be related to its function, while the variability between ensembles in the super-ensemble quantifies the uncertainty of the model. Such a representation is implemented, for example, in the ASTEROIDS software [42]. It provides a straightforward way for predicting any observable quantity, including its uncertainty. Current Bayesian approaches provide a single set of conformer weights that include both uncertainty and ensemble width.

## 6. Specifics of Label-Based Restraints

Label-based EPR and fluorescence techniques provide valuable experimental restraints in a distance range inaccessible to most other techniques which are applicable to partially disordered systems. Compared to small-angle scattering, they can provide more detailed information with higher resolution. This benefit comes at the cost of a relatively high sample preparation cost and a low yield of restraints per sample. Therefore, integrative or hybrid approaches to modelling are used in most applications. An additional problem arises from the introduction of the labels, whose size and conformational distribution exceed the accuracy of the measurements [16,55]. In order to take full advantage of label-based restraints, the labels must be considered in the modelling [56].

### 6.1. Label Position versus Backbone Position

DEER depends on the distance between the spin density centres of two paramagnetic markers, while smFRET depends on the distance between the donor and acceptor chromophores. The latter is usually assumed to be the distance between the chromophore centers. In modelling, one usually tries to restrain the distance between backbone atoms, e.g., between Cα atoms in proteins. The positions of the labels differ from the position of the representative backbone atom of the labelled residue by at least a few Angstroms. Moreover, the distance $r_{\mathrm{bl}}$ between the backbone and the label is distributed due to the conformational distribution of a flexible linker. Flexible linkers are used to minimise the risk of perturbing the conformational distribution of the backbone. In the case of smFRET, a flexible linker is also required to achieve sufficient orientation averaging of the factor κ related to the direction of the transition dipole moments of the donor and acceptor with respect to the distance vector [57]. The spatial distribution of the labels matters even if one restrains only the mean distances, because the magnetic dipole-dipole interaction measured with DEER EPR involves $r^{-3}$ ensemble averaging and the smFRET efficiency involves $r^{-6}$ ensemble averaging. Therefore, the spatial distribution of the label position should be taken into account at least in the ensemble refinement (reweighting). In contrast, when generating a raw ensemble, it may be advantageous to consider only an approximate mean position of the labels. Otherwise, modelling the labels may become the time-limiting step of the conformer generation.

### 6.2. Representation of Labels by a Rotamer Library

The distance vector ${\vec{r}}_{\mathrm{bl}}$ depends on the bond lengths, bond angles and torsion angles $\chi_{i}$ of the flexible linker and of the chromophore or free radical. The distributions of bond lengths and bond angles are comparatively narrow and the chromophores and free radicals usually do not contain rotatable bonds. Therefore, modelling ${\vec{r}}_{\mathrm{bl}}$ and its distribution amounts to modelling the correlated distributions of the $\chi_{i}$. This problem is equivalent to modelling native amino acid side chains [58] and can be solved by a rotamer library approach [59,60,61]. In this approach, the distribution of torsion angles $\chi_{i}$ is discretised. Each set of torsion angles for the free label can be assigned a population. The populations are computed by projecting a raw ensemble, which can be obtained by MD simulation [60] or Monte Carlo sampling [61], onto the discrete set of canonical rotamers. For most spin labels, the number of rotatable bonds ranges from 4 to 6 and the number of discrete states per rotatable bond ranges from 2 to 6, while some rotamers are excluded due to steric constraints. This leads to comprehensive libraries with about 100 to 1000 rotamers. Figure 6a shows example for the rotamer representation of a spin label at a surface-exposed site.

Chromophores are larger and their linkers typically include between 8 and 13 rotatable bonds. Since the number of canonical rotamers grows exponentially with the number of rotatable bonds, it is not possible to use comprehensive libraries. For 10 or more rotatable bonds, even generating a raw ensemble by exhaustive sampling and its projection onto discrete states by hierarchical clustering become a challenge. On the other hand, smaller reduced sets of no more than 1024 representative rotamers are sufficient to sample the accessible space [61].

Rotamer libraries allow the consideration of the interaction between the label and the macromolecule through a statistical thermodynamics approach [59]. To this end, MMM [60] calculates the interaction energy through a simplified atomistic force field. In the simplest version, electrostatic interactions are neglected and a Lennard-Jones potential is used to account for repulsion (avoiding collisions between label and macromolecule) and van der Waals attraction. The interaction energy is added to a free label energy obtained by Boltzmann inversion of the rotamer populations in the library, and a new Boltzmann distribution of the populations is calculated from the energies of the attached label. Rotamers with very small populations, typically accounting for 1% of the total population, are discarded. Although such a discretisation and approximation for the interaction potential may seem crude, it proves difficult to obtain more accurate models, even with full MD approaches that require orders of magnitude more computational effort [63]. In the rotamer library approach, the Lennard-Jones potential of a standard molecular force field is scaled down by a “forgive” factor that compensates for the narrowing of the linker conformational distribution by discretisation and presumably also for the shielding of the attractive part by solvation. For chromophores [61] or stickier labels [64,65], scaling down the Lennard-Jones potential by a “forgive” factor is also required in rotamer library construction.

Another significant improvement in computational efficiency without reducing accuracy was achieved in the RosettaEPR context by replacing the label with a single pseudoatom. The pseudoatoms calculated from the original rotamer library of this package were fitted to an extensive set of experimental DEER data for T4 lysozyme and the redundancy of the discrete set of pseudoatoms was reduced [63]. This representation is visualised in Figure 6c.

In the Pronox approach, a library of preferred rotamers is created based on label conformers observed in X-ray crystal structures of labeled proteins [66]. All of these rotamers are assigned a population of 0.9, while other sterically possible rotamers are assigned a population of 0.1. The interaction with the protein is based on the binary collision criterion, in which some of the rotamers are discarded. Although populations of label pairs can be obtained in this way with less computational effort, the overall approach is somewhat slower than that of MMM, presumably because it involves a fine search for possible conformations upon binding to the protein. The accuracy of the predictions is at the level of MMM [67].

### 6.3. Representation of Labels by Accessible Volume

Similar to the pseudoatom approach in RosettaEPR, the accessible volume approaches abstract partially or even fully from the internal structure of the label. The most popular of these approaches in EPR is mtsslWizard [68,69], which scans the distribution of torsion angles $\chi_{i}$ neglecting the torsion angle potential. The generated label conformers are checked for internal collisions and collisions with the protein using a binary collision criterion for which the user has two choices. In many cases, mtsslWizard performs equally well as MMM and Pronox [67], albeit with slightly greater computational effort [63], presumably because it scans a much larger number of conformers.

An even simpler description of the label is used in an accessible volume approach used for FRET chromophores [57,70]. In this approach, the chromophore is represented by a sphere of radius $R_{\mathrm{dye}}$ and the linker by a cylinder of effective length $L_{\mathrm{link}}$ and effective width $w_{\mathrm{link}}$ (Figure 6b). The distribution of the label position is obtained by varying the orientation of the linker with respect to the backbone and discarding all orientations that cause collisions with the macromolecular surface. The three geometric parameters $R_{\mathrm{dye}}$, $L_{\mathrm{link}}$ and $w_{\mathrm{link}}$ depend on the structure of the label. The authors assumed a common linker width of 4.5 Å and estimated $L_{\mathrm{link}}$ from the fully extended conformation of the linker. For RNA labelling, this model was found to be oversimplified when run with a single value of $R_{\mathrm{dye}}$; a problem that can be remedied by running it for three different radii [57]. A recent study seems to indicate that the accuracy of the predicted smFRET efficiencies is slightly better with a rotamer library approach than with the accessible volume approach [61]. However, a larger set of comparative simulations would be needed to draw a firm conclusion.

### 6.4. Representation of Labels in MD Simulations

MD simulations of spin-labelled proteins can be performed at a level where they almost perfectly predict sets of continuous-wave EPR spectra recorded at multiple microwave frequencies [71]. Because such sets of spectra are very sensitive to the time scale and specifics of side-chain motion and are notoriously difficult to fit with simple models for dynamics, this suggests a quantitative match between the simulated and real side-chain dynamics of spin labels. However, there are some caveats. First, the approach has been tested for surface-exposed spin labels at easily accessible sites. It may not work as well for the slower dynamics and stronger label-protein interactions in denser label environments, which cannot always be avoided. Second, the transitions between some of the rotameric states of the spin-label side chain are so slow that a large number of very long MD trajectories would be required for sufficient sampling. The latter problem can be solved by constructing a Markov state model whose transition matrix quantifies the jump rates between rotamer states [72], albeit at considerable expense that may be unrealistic in the context of hybrid ensemble modeling. Therefore, one resorts to less sophisticated approximate representations.

The conceptually simplest representation uses multiple replicas of the spin label to improve the sampling of rotamer states [73]. This still involves a significant computational cost. In a simplified model, the position of the spin is represented by a dummy ON particle positioned with respect to the N, Cα, and Cβ atoms by parameterized force field terms [62]. Of course, this parameterization abstracts from the local packing density. Nevertheless, the evaluation for 37 surface-exposed sites in T4 lysozyme gave a slightly better agreement with experiment than with a static structure and the original rotamer library in MMM [62].

It is also possible to perform MD simulations for the native system and then calculate the conformational distribution of the spin label for as many trajectory frames as needed. For this purpose, one can use the rotamer library approach [74,75]. This approach facilitates the addition of new labeling sites, but cannot be applied when the distances between labels are used to constrain MD simulations. A similar approach with an accessible volume model has been proposed for smFRET chromophores [70].

### 6.5. Improving Label Representation with Experimental Information

The rotamer library and accessible volume approaches predict mean label-to-label distances with an uncertainty of about 3 Å (standard deviation) [67,68]. The situation does not improve substantially with more elaborate MD simulations [62]. Such accuracy is sufficient in cases where substantial disorder limits the resolution of ensemble models, but disappointing in cases where the conformation is very well defined. Although a well-defined structure in the Anfinsen limit (Figure 2a) can be better characterised using classical structural biology approaches, there are situations where one may want to resort to label-based techniques. This is the case, for example, when a protein can be crystallised in one of its functional states but not in another. The structure of the latter state can then be elucidated with labelling-based restraints, albeit only with accuracy and resolution limited by the quality of the spin-labelling representation. The same problem arises in the elucidation of differences between a known crystal structure and an unknown solution structure of a system.

This problem can be addressed by rotamer reweighting in terms of the Bayesian approach BioEn [46]. The method was implemented by fitting to DEER time-domain data, avoiding explicit calculation of distance distributions. For the three N-terminal polypeptide transport-associated (POTRA) domains of Omp85, modest changes in rotamer weights resulted in near-perfect agreement with experiment already for the crystal structure [75]. A very small further improvement was achieved by shifting the domains by 1–3 Å. A similar approach implemented in RosettaEPR resulted in improved modeling of a conformational change of the transition from the outward to the inward state of the multidrug transporter PfMATE [76].

### 6.6. Influence of Label Dynamics

The label conformation is not only distributed, but also dynamic. While DEER distance distributions are measured at temperatures between 50 and 80 K, where these dynamics are stalled on the time scale of the experiment, smFRET data acquisition overlaps with the time scale of side-chain dynamics at ambient temperatures. The problem can be addressed with reasonable computational effort by modeling the dynamics of the labels as diffusive motion in a potential of mean force [77]. In a recent study, this approach was found to improve the prediction of smFRET efficiencies for both representations of the conformational distribution of the label by accessible volume and rotamer library approaches [61].

## 7. Specifics of Distribution Restraints

Most experimental techniques provide data either as a list of expectation values of observables with their standard deviations or as a list of lower and upper bounds. In either case, prediction based on an ensemble model yields a single value corresponding to the ensemble average. The corresponding value for a single conformer, in the absence of knowledge about other possible conformers, does not tell us whether that conformer is consistent with the data. The situation is different for distribution restraints. The distribution assigns a probability to the number predicted for a single conformer. Given a list of independent distributional restraints, the individual (marginal) probabilities multiply to a joint probability. Conformers whose joint probability is below a certain threshold can be discarded without knowing the other potential conformers.

Distributional restraints also behave differently when reweighting ensembles. In extreme cases, all observable values predicted for a single conformer may match the expectation values of a broad ensemble within their uncertainty. In such a case, mean-value restraints would suggest perfect order, although the system may be substantially disordered. If discrepancies occur for predictions for a single conformer, it is still likely that they can be resolved by assuming an ensemble much narrower than the actual distribution of conformers. This is one reason why mean-value restraints should be supplemented with distributional restraints, even in cases where the number of mean-value restraints is sufficient for deriving a model of the system.

In the following, we discuss these issues for distance distribution restraints that can be obtained by DEER EPR. Analogous considerations apply to angular distribution restraints between three sites and dihedral angular distribution restraints between four sites, which may become experimentally accessible in the future.

### 7.1. Should We Use Primary Data or Spatial Restraints for Model Evaluation?

An ensemble model allows us to easily compute the probability distribution $\vec{P}$ of any geometric parameter, which we consider here to be discretized so that it can be represented by a vector. Predicting the form factor $\vec{F}$ (Figure 7c) from $\vec{P}$ (Figure 7b is a mathematically well-posed problem that relies on multiplying a kernel matrix K by the distribution $\vec{F}=K\vec{P}$. The uncertainty of the prediction arises only from the uncertainty of the kernel, which in turn results from the fact that experimental parameters or further sample parameters are not precisely known, or that the physics of the experiment is incompletely described. In simulation of the primary data (Figure 7a), the main errors arise from the uncertainty of the spin concentration and the uncertainty of the inversion efficiency of the pump pulses that determine the background.

In contrast, computing $\vec{P}$ (Figure 7b) from $\vec{V}$ (Figure 7a) [16] is a mathematically ill-posed problem, since the kernel matrix K has a high condition number. Therefore, inversion of K leads to solutions that are unstable even for small deviations of $\vec{V}$ from the physical model. This problem can be mitigated by representing $\vec{P}$ by a parametric model with few parameters, by regularization, or by neural network analysis [16]. The simplest model, a single Gaussian (Figure 7d), usually leads to a stable fit, but may suppress significant details of the distribution. In general, converting $\vec{V}$ to $\vec{P}$ increases uncertainty compared to converting $\vec{P}$ to $\vec{V}$. Therefore, a direct fit to the primary $\vec{V}$ data can reduce the uncertainty. This strategy has been used for high-resolution docking of protomers in the dimer of the sodium/proton antiporter NhaA [78] and in high-resolution approaches involving reweighting of rotamer populations [75,76]. With such approaches, a good estimate for the distance distribution is available during background fitting. This allows the use of DEER EPR data with shorter trace length [63,75].

Direct fitting of primary data also has drawbacks. Some algorithms, such as deformation of an elastic network model by distance-dependent forces [79], and some modelling platforms, such as IMP [80], require explicit spatial restraints. In addition, the need for background adjustment during model scoring can become a computational bottleneck when scores need to be computed for a very large number of models. This situation occurs, for example, when reweighting large raw ensembles. In any case, it may be advisable to calculate and visually check the distance distributions and their uncertainties, even if primary data are used for model evaluation. This is because assessing data quality in the distance domain is easier and more intuitive.

### 7.2. Distributional Restraints in Raw Ensemble Generation

In generating raw ensembles, an attempt is made to draw a sample from the entire conformational space whose statistics are as close as possible to the conformer statistics of the system under study. The sample size may be chosen larger than the expected size of a representative ensemble to allow more flexibility in reweighting the ensemble. Monte Carlo approaches, MD simulations, or algorithms specific to a particular modeling task are used for sampling. The sampling of rigid body arrangements in MMM [81] and MMMx [41] is an example of such a specific algorithm. The relative spatial arrangement of *n* rigid bodies is fully specified by 3(n−1) translation parameters and 3(n−1) Euler angles. By assigning 3 reference sites in each rigid body, 9m(n−1)/2 pairs of sites located in different rigid bodies are obtained. The set of distance pairs overdetermines the relative spatial arrangement, and the solution can be computed directly by distance geometry. If distance distributions are available, exhaustive sampling is possible by setting lower and upper bounds for each distance and defining a small set of distance samples between these bounds (for example, see purple lines in (Figure 7d)) [81]. This notion of lower and upper bounds is different from the notion used for mean-value restraints. For mean values, the bounds are a measure of the uncertainty of the restraint, while for distributions they are a measure of the distributional width.

Monte Carlo sampling for peptide chains can be based on Ramachandran statistics, preferably residue-specific Ramachandran statistics for peptide segments without secondary structure [39,40]. Additional secondary structure propensities can be defined. Flexible Meccano can be used to specify that long distance contacts must be satisfied for a certain proportion of conformers. Distance distribution restraints can be evaluated in MMM and MMMx once both residues of a pair of spin-labelled sites have been generated. Such an evaluation provides a probability that the conformer is part of the intended statistical sample. As soon as this probability falls below a threshold, the conformers can be discarded [39]. Since unrealistic conformers do not need to be fully computed, this strategy improves sampling efficiency.

In MD simulations, distributional restraints can be introduced by representing them through a histogram in the form of Gaussian functions with a standard deviation of 1.7 Å [73]. This representation provides smooth analytical derivatives for the restraining potential. An empirical force constant of 10,000 (kcal/mol)/Å${}^{2}$ was used to ensure a good fit of the distribution restraints. Working with this approach is made easy by an auxiliary package for CHARMM, which includes a graphical user interface [82].

Alternatively, a pseudo-energy term can be derived by interpreting the distance distribution as a Boltzmann distribution generated by a potential of mean force,
(1)$$f=k_{B}T\frac{d\ln P(r)}{dr}\;.$$

It is convenient to express P(r) as a sum over functions whose logarithm can be differentiated by *r*. This choice balances the distance distribution restraints with the energy terms of the molecular force field, rather than strictly enforcing them.

### 7.3. Distribution Restraints in Ensemble Reweighting

Ensemble reweighting requires finding a balance between different experimental methods and, in hybrid approaches, between experimental restraints and the molecular force field. The optimal balance can only be determined if the uncertainties of all experimental restraints and the force field are known. In Bayesian approaches, only statistical errors of the experimental data are considered in terms of $\chi^{2}$ values, and the error of the force field is assumed to be as small as it can possibly be given the experimental restraints and their uncertainty. Since the calculation of distance distributions from experimental data is an ill-posed problem, $\chi^{2}$ is not necessarily a good metric for such distributions. Bayesian approaches to ensemble reweighting are therefore better combined with fitting to primary data [75], for which $\chi^{2}$ is well-behaved.

Artefacts in P(r), such as those that can arise from under-regulation or the presence of narrow peaks with high amplitude when fitting multi-Gaussian distributions, have less effect when the overlap of the distributions is used as a metric [18]. The overlap is defined as the joint area of two distribution functions $P_{\mathrm{sim}}\left(r\right)$ and $P_{\exp}\left(r\right)$, where each of the functions has a total area of one. Ensemble reweighting with only distance distribution restraints determines the set of conformal weights that minimises the overlap deficit, i.e., the deviation of the overlap from unity. The overlap values are geometrically averaged so that a very small overlap in individual restraints is heavily penalised [81].

In many situations, the uncertainty of the distance distribution restraints is dominated by the uncertainty of the representation of the spin labels. If only the experimental uncertainty of the distribution is considered, this may lead to an inappropriate weighting of these restraints. Furthermore, since the distance distribution restraints are measured in glassy frozen samples and since the labels may distort the structure, these restraints can be inconsistent to some extent with other experimental restraints. Such partial inconsistency of experimental restraints from different techniques is a problem that can also occur with other data. Therefore, it may be useful to determine the uncertainty and consistency of the experimental data set during ensemble reweighting. For this purpose, the ensemble reweighting is first performed for each individual subset of homogeneous experimental restraints, i.e., for subsets of restraints originating from the same technique. In this way, a set of figure of merits $m_{s,1}$ is obtained, indicating how well the reweighting of a given raw ensemble can fit the data of each individual experimental technique. The figure of merit is defined such that its minimum corresponds to the best fit of the experimental data. In a second step, all restraint subsets are considered simultaneously, resulting in a set of conformer weight dependent figures of merit $m_{s,2}$. In contrast, the $m_{s,1}$ are fixed in the second reweighting. One now minimises the loss of merit,
(2)$$L=\left(\frac{1}{S}\sum_{s=1}^{s}\frac{m_{s,2}}{m_{s,1}}\right)-1\;,$$
where *S* is the number of subsets of restraints from different experimental techniques. The loss of merit *L* is a non-negative number, where L=0 means a complete match between all restraint subsets. For L=1, the simultaneous matching of all subsets of restraints causes the figures of merit to double on average. Values L>1 indicate significant inconsistencies that should be resolved or at least explained before accepting the ensemble model.

Figure 8 illustrates the effect of ensemble reweighting by 19 distance distribution restraints and a SAXS curve for the free, dispersed form of the RNA-binding protein hnRNP A1 [21]. In this case, reweighting combined with discarding conformers with less than 1% of the population of the most populated conformer reduces the number of conformers from 1119 (raw ensemble in Figure 8a) to 138 (reweighted ensemble in panel (b)) while improving restraint fits as compared to the raw ensemble with uniform populations. Note that in this case the 19 distance distribution restraints were already considered in generation of the raw ensemble.

## 8. Outlook

To date, ensemble modelling with label-based restraints has been pursued by only a few research groups. It is the rule rather than the exception that a new application requires further method development. Software packages exist for FRET-based labeling (FRET Positioning System FPS 2.0, [83]) and for DEER EPR-based modelling (Multiscale Modelling of Macromolecules extended, MMMx, [41]). Both packages allow integration with high-resolution structures obtained by other methods, and MMMx supports integration with small-angle scattering and NMR-PRE data. When generating raw ensembles, MMMx can use secondary structure propensities derived from NMR experiments. Future developments should allow integration of more experimental data, especially at the ensemble reweighting stage.

MD simulations can be restrained by distance distributions using the CHARMM-GUI DEER facilitator [82], which can also compute spin pair distance distributions from MD trajectories. For the latter task, one can also use the DEER-PREdict package [74] or MMMx [41]. MMMx and BioEn [75] allow ensemble reweighting with distance distribution restraints. Current approaches to distribution-based modelling provide ensembles without uncertainty estimates, i.e., ensemble width and uncertainty are not separated. A possible, albeit computationally intensive, solution to this problem is jackknife resampling [18] or bootstrapping. Less computationally intensive approaches should be explored for estimating the uncertainty of the ensemble model or of the expectation values of observables.

In the context of uncertainty estimation, it is currently difficult to predict how many experimental restraints are required to obtain a reliable ensemble model. Usually, this question can only be answered in the context of a validation. If the result of such a validation is not satisfactory or inconclusive, further experimental restraints have to be acquired. This strategy would benefit from a better understanding of how the different types of restraints best complement each other. This issue is particularly pressing for hybrid approaches because there is no systematic approach to estimating uncertainty in MD simulations.

In summary, integrative and hybrid structure determination based on label-based long-range distance restraints and distance distribution distance restraints has gained importance in recent years. The first ensemble models of biological systems of current interest have been created, showing why partial disorder can benefit function. Further developments in this direction may be our best chance to characterize systems that lie between the extremes of perfect order and total disorder.

## Figures and Tables

**Figure 1 biomolecules-12-01369-f001:**
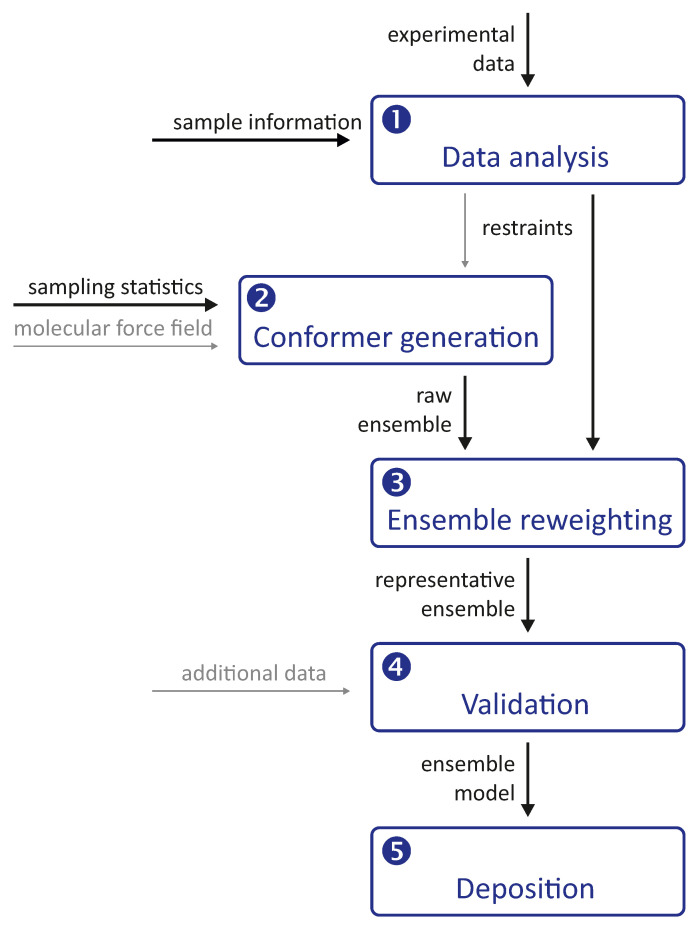
General pipeline for integrative and hybrid ensemble modelling. Thick black arrows denote required inputs, thin grey arrows denote optional inputs. The pipeline is preceded by the design and execution of experiments and is complemented by ensemble analysis. Numbers 1 to 5 denote the main modules that can be implemented using different approaches.

**Figure 2 biomolecules-12-01369-f002:**
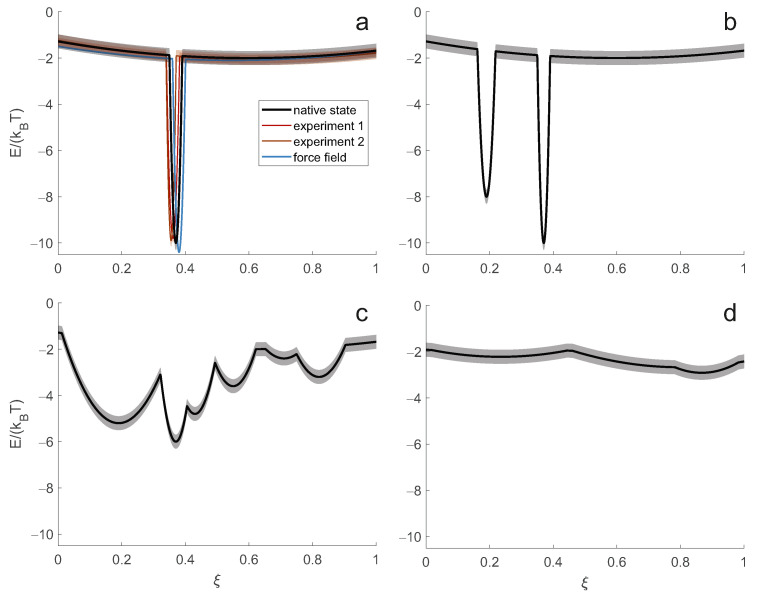
Classification of energy landscapes of biomolecular systems. Energy landscapes are associated with uncertainties (semitransparent bands), as explained in the text. (**a**) Anfinsen limit with a single minimum several times the thermal energy $k_{B}T$ lower than the rest of the hypersurface. Shown are the slightly different energy landscapes corresponding to the system in the living cell (black), two different experimental series 1 (red) and 2 (orange-red), and a molecular force field (blue). (**b**) System with two separate minima differing by only a small multiple of $k_{B}T$. (**c**) Energy landscape, with ruggedness on the order of $k_{B}T$. (**d**) Energy landscape that is nearly flat on the order of $k_{B}T$.

**Figure 3 biomolecules-12-01369-f003:**
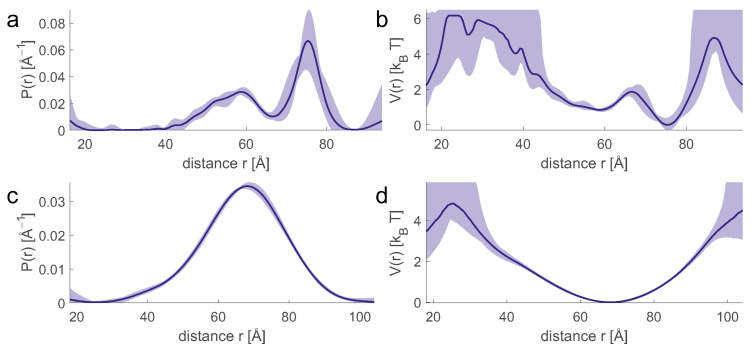
Equivalence between distance distributions (**a**,**c**) and potentials of mean force (**b**,**d**) for protein site pairs 71/475 (**a**,**b**) and 235/475 (**c**,**d**) in the complex of polypyrimidine tract binding protein 1 with encephalomyocarditis virus internal ribosome entry site. Data from [31].

**Figure 4 biomolecules-12-01369-f004:**
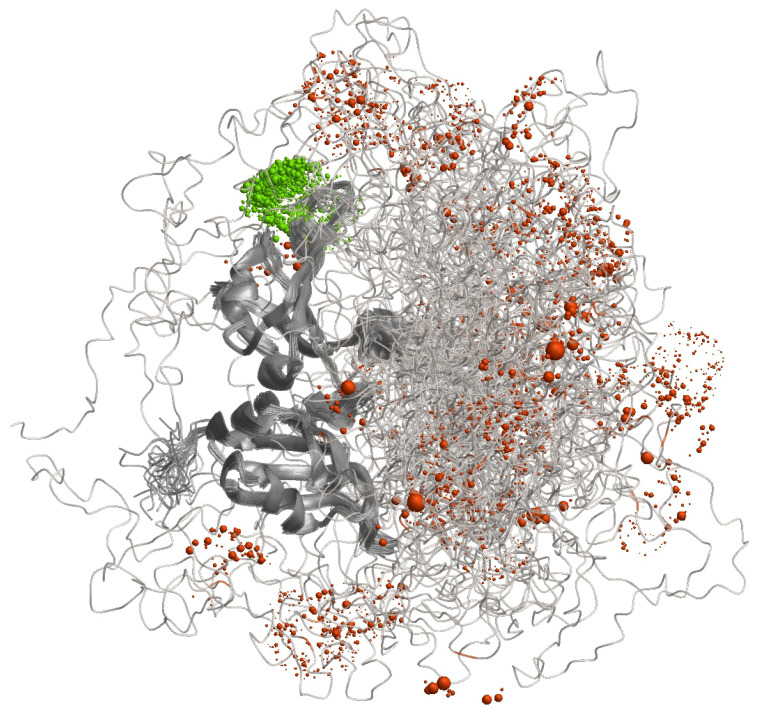
Characterisation of conformer distribution by labelling approaches. Shown is an ensemble model for the RNA-binding protein hnRNP A1 in its free, dispersed form (model determined with DEER and SAXS restraints from [21]). The ensemble was reduced to 46 conformers for clarity of display. Conformers are superimposed on the two RNA-binding domains (residues 1–186). The glycine-rich N-terminal domain (residues 187–320) is largely, but not completely disordered. A spin label at site 144 in the ordered domain (green) is narrowly distributed in space. A spin label at site 252 in the disordered domain (orangered) is broadly distributed in space. Computation of spin label positions by a rotamer library and visualization were performed by MMM.

**Figure 5 biomolecules-12-01369-f005:**
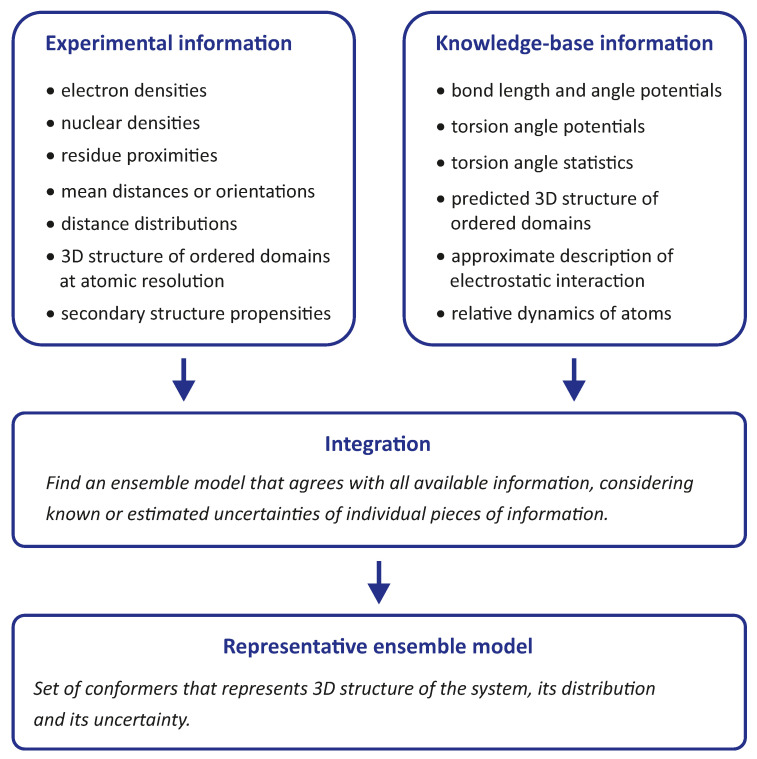
Integration of experimental information and information from knowledge bases into a hybrid ensemble model. The lists of information types for specifying ensembles are examples rather than being exhaustive. The balancing of potentially inconsistent information in the integration step is complicated by partially unknown uncertainty of pieces of information. Construction of a representative ensemble model is complicated by the requirement of separation of uncertainty from natural distribution of 3D structure.

**Figure 6 biomolecules-12-01369-f006:**
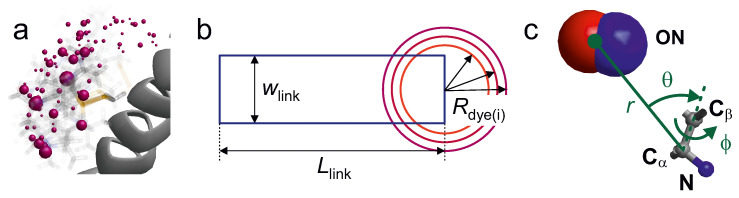
Representation of labels in modelling approaches (**a**) Rotamer library representation (MMM). Rotamer population is encoded by transparency and by the volume of the purple spheres that represent the N-O group midpoint of rotamers. (**b**) Accessible volume model parametrized by a linker length $L_{\mathrm{link}}$, a linker width $w_{\mathrm{link}}$, and a set of three dye radii $R_{\mathrm{dye}(i)}$. Adapted from [57] (**c**) Coarse-grained rotamer model based on a dummy ON particle, which represents the midpoint of the N-O group. Each rotamer is defined by a distance *r* from the Cα atom and two angles that relate the label position to the Cα-Cβ bond. Adapted from [62].

**Figure 7 biomolecules-12-01369-f007:**
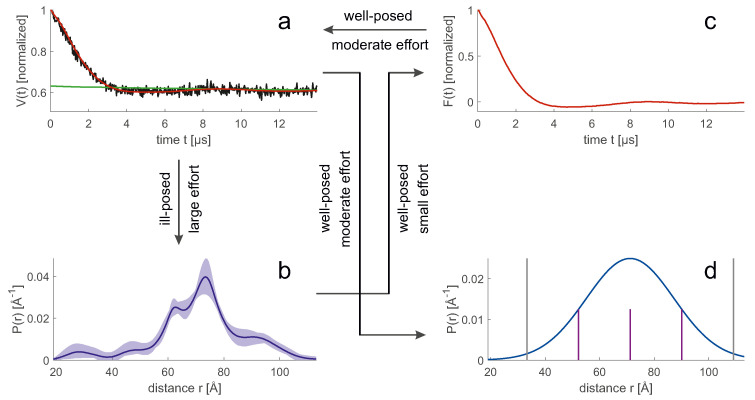
Different representations of a distance distribution in modelling. Data corresponds to spin-labelled site pair 202/475 in the complex of polypyrimidine tract binding protein 1 with encephalomyocarditis virus internal ribosome entry site [31]. Transformations between representations can be well-posed (stable) or ill-posed (potentially unstable) and may require different computational effort. (**a**) Primary DEER EPR data (black), intermolecular background (green), and fit by a distance distribution (red). (**b**) Model-free distance distribution (blue) with 95% confidence interval (pale blue). (**c**) Form factor that results from separation of the label-pair contribution from intermolecular background. The form factor is fully determined by the distance distribution. A fit to primary data involves optimisation of background parameters. (**d**) Gaussian distribution, which is fully determined by a mean value 〈r〉 and standard deviation $\sigma_{r}$ (blue). Grey vertical lines denote twice the full width at half maximum. Purple vertical lines denote three equidistant distance samples used for exhaustive discrete sampling.

**Figure 8 biomolecules-12-01369-f008:**
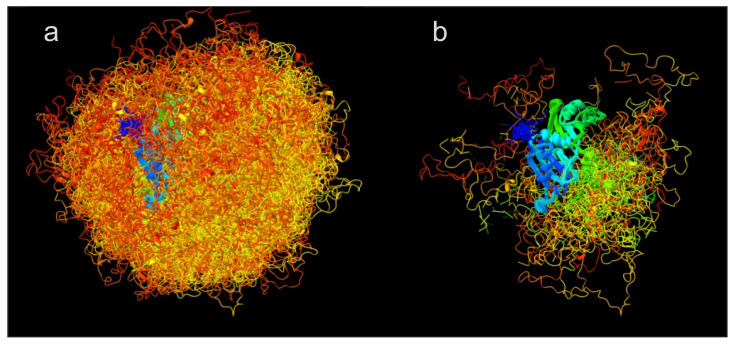
Effect of distance distribution restraints in ensemble reweighting. Shown are the raw ensemble (**a**) consisting of 1119 conformers for the RNA-binding protein hnRNP A1 in its free, dispersed form ([18]) and the reweighted ensemble (**b**) of 138 conformers from integrating information from 19 DEER distance distribution restraints and a SAXS curve ([21]). Visualization was performed by ChimeraX.

**Table 1 biomolecules-12-01369-t001:** Selected software packages for integrative modelling and ensemble modelling with label-based restraints.

Package	Purpose	URL^1^
ASTEROIDS	Ensemble reweighting	tinyurl.com/2fhuf4wj ^2^
BioEn	Ensemble reweighting	github.com/bio-phys/BioEn
DEER-PREdict ^3^	Distance distributions and PRE	github.com/KULL-Centre/DEERpredict
Flexible Meccano	Raw ensemble	tinyurl.com/2fhuf4wj
FPS	smFRET label modelling	github.com/Fluorescence-Tools/FPS
FRETrest ^4^	restrained MD	github.com/Fluorescence-Tools
IMP	Modelling pipeline	integrativemodeling.org/
MMMx	Modelling pipeline	github.com/gjeschke/MMMx
mtsslSuite ^5^	Distance Distributions	mtsslsuite.isb.ukbonn.de
PLUMED-ISDB	Modelling pipeline	plumed.org
reMD Prepper ^6^	restrained MD	charmm-gui.org
Spin-Pair Distributor ^6^	Distance distributions	charmm-gui.org
Yasara ^7^	Structure refinement	yasara.org

^1^ All URL accessed on 24 September 2022. ^2^ in preparation. ^3^ Python package: pip install DEERPREdict. ^4^ Requires proprietary AMBER. ^5^ Server version including modules for several tasks. ^6^
Requires proprietary CHARMM. ^7^ Proprietary software.

## Data Availability

Not applicable.

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
