# Peer review of "Integration of Nanometer-Range Label-to-Label Distances and Their Distributions into Modelling Approaches"

_biomolecules, 2022, doi:10.3390/biom12101369_

Round 1
Reviewer 1 Report
This is a clear and thorough review of current methods of biomolecular ensemble modeling. The review maintains the intended focus on distance and distance distribution restraints derived from DEER EPR and smFRET while providing context and useful information about complementary methods. The review also includes a description of the statistical thermodynamics underlying ensemble modeling that is valuable for understanding the problem and the various solutions.
The review will be an excellent resource for current practitioners and those planning to use integrative and hybrid approaches to biomolecular ensemble modeling as well as those seeking background to aid in the interpretation of models generated using these methods.
Author Response
I thank the reviewer for reading and assessing the manuscript and I am happy with the positive judgement.
Reviewer 2 Report
The text and figures on the manuscript lack detailed descriptions of specific examples. I cannot regard the manuscript as a review. A review article should not be a lengthy description of a concept, but should be clearly written with concrete examples. As it stands, this manuscript is not worthy of publication.
Author Response
I thank the reviewer for reading the manuscript. Given the contrary opinions of the two other reviewers, I will not change the concept of my review. In my opinion, an exposition of underlying concepts with references to their in-depth description is more valuable than a collection of concrete examples. I do not review applications here, but the development of methodology.
Reviewer 3 Report
There is a lack of knowledge on the most effective ways to model protein conformational heterogeneity and disorder. This is particularly complex given the understanding that proteins and domains of proteins may occupy a continuum of conformations.
This review has focussed on approaches to ensemble modelling of these conformational heterogeneities from nanoscale distance based experimental data, particularly DEER EPR and smFRET.
This review provides a highly comprehensive, informative and detailed overview of the current state of approaches of modelling distances between labels on proteins in order to gain full conformational information. I have very few comments and the review is certainly fine to publish in its current state, I believe it could benefit from the inclusion of additional schematic figures to better illustrate more complex points, i.e. a conceptual figure illustrating a labelled protein/protein domain early in the review would be beneficial. I think this review has the potential to benefit others that may work on the periphery of this field, such as single-molecule microscopists and biophysicists that do not perform modelling themselves, the inclusion of additional conceptual figures would help with this.
Author Response
Thanks for your thoughtful report. I agree that the ratio between text and figures was not optimal.
- I added the requested "conceptual figure illustrating a labelled protein/protein domain" (new Figure 4). This new figure is based on a recently published ensemble structure of hnRNP A1. This example allows for illustrating spin label distribution in an ordered and in a disordered domain in the same protein.
- I added a schematic figure on types of information that are integrated, which includes (hopefully) meaningful definitions of integration and of a representative ensemble model (new Figure 5).
- Finally, I added a figure that illustrates the effect of ensemble reweighting with distance distribution (and SAXS) restraints (new Figure 8). This figure is based on the same hnRNP A1 model as used in new Figure 4 as well as on the raw ensemble that underlies this model.
I am aware that new figures 4 and 8 are a bit more illustrative than conceptual, but I hope that these illustrations are helpful for grasping the concepts explained in the text.